# Optimization and Evaluation for the Capacitive Deionization Process of Wastewater Reuse in Combined Cycle Power Plants

**DOI:** 10.3390/membranes13030316

**Published:** 2023-03-09

**Authors:** Yesol Kim, Hyeongrak Cho, Yongjun Choi, Jaewuk Koo, Sangho Lee

**Affiliations:** 1School of Civil and Environmental Engineering, Kookimin University, 77 Jeongneung-ro, Seongbuk-gu, Seoul 02707, Republic of Korea; 2Korea Institute of Civil Engineering and Building Technology, 283 Goyang-daero, Ilsanseo-gu, Goyang-si 10223, Republic of Korea; 3Desalination Technologies Research Institute (DTRI), Saline Water Conversion Corporation (SWCC), P.O. Box WQ36+XJP, Al Jubayl 35417, Saudi Arabia

**Keywords:** combined cycle power plant, water reuse, capacitive deionization, optimization, fouling, cleaning, response surface methodology

## Abstract

Combined cycle power plants (CCPPs) use large amounts of water withdrawn from nearby rivers and generate wastewater containing ions and pollutants. Despite the need for wastewater reclamation, few technologies can successfully convert the wastewater into make-up water for CCPPs. Therefore, this study aimed to apply capacitive deionization (CDI) for wastewater reclamation in CCPPs. Using a bench-scale experimental unit, which included ion exchange membranes and carbon electrodes, response surface methodology (RSM) was used to optimize the operating conditions of the CDI process to increase the total dissolved solids (TDS) removal and product water ratio. The optimal conditions were found to be a voltage of 1.5 V, a flow rate of 15 mL/min, and an adsorption/desorption ratio of 1:0.8. The changes in CDI performance with time were also studied, and the foulants on the membranes, spacers, and electrodes were examined to understand the fouling mechanism. The TDS removal decreased from 93.65% to 55.70% after 10 days of operation due to the deposition of scale and organic matter. After chemical cleaning, the TDS removal rate recovered to 93.02%, which is close to the initial condition.

## 1. Introduction

Combined cycle power plants (CCPPs) account for over 80% of the world’s electricity generation. Most of these plants use electric steam technology, which uses large amounts of water to produce steam that turns turbines to generate electricity. The blowdown from the steam cycle is typically quenched to 60 °C. To recycle this stream, a heat exchanger or chiller must be installed in the recycle path for reuse [1]. Combined heat and power plants combine two or more thermal cycles to increase the efficiency of energy use. A representative model is a mixture of a gas turbine cycle and a steam turbine cycle [2,3]. There are two major challenges with CCPPs. First, they use a large amount of pure water to operate the generator turbine and boiler. They also require cooling water to reduce the water temperature inside the plants [4,5]. Second, CCPPs generate wastewater that can harm the environment and human health if not properly treated. The high concentration of organic matter and hardness in the CCPP wastewater remaining in the operating system are the main causes of scale and fouling. This not only reduces the performance of the plant, but also degrades the quality of the treated water from the CCPPs and reduces the quantity of production. In addition, if it is used for irrigation, it can harm crops and make the soil less fertile [6,7,8]. As a result, discharge regulations for treated wastewater from CCPPs have become more stringent.

To address this issue, CCPPs must implement effective wastewater treatment and disposal strategies. A common approach is to treat the effluent using physical, chemical and biological processes to remove contaminants and produce clean water that can meet discharge regulations. These include sedimentation, coagulation and flocculation, activated sludge, or reverse osmosis. Another approach is to completely recycle and reuse wastewater after proper treatment, also known as zero liquid discharge (ZLD). Unlike the first approach, ZLD is free from environmental regulations because it allows for the complete elimination of pollutants. It also reduces water consumption, making it sustainable and environmentally friendly [9,10,11].

However, the application of ZLD to CCPPs is challenging due to the lack of appropriate and affordable technologies [8,12,13,14]. To reuse the wastewater, it is essential to remove ionic species to meet the process water quality. Conventional technologies include reverse osmosis (RO), electrodialysis (EDI), and mechanical vapor recompression (MVR) [15,16,17,18,19,20], which have drawbacks such as high energy consumption and high potential for scale and fouling [21,22]. Capacitive deionization (CDI) is an alternative to conventional deionization because it can effectively remove a wide range of ions, including dissolved salts, heavy metals, and other contaminants [23,24]. CDI requires relatively low energy consumption and maintenance, making it a cost-effective option [25,26,27]. Because of these advantages, CDI has been used to treat feed water that contains less than 2.0 g/L of total dissolved solids (TDS) [28].

Although CDI has the potential for ZLD in CCPPs, there are several issues to consider when applying CDI to the CCPP effluent [29]. Although the fouling potential in CDI is lower than in conventional pressure-driven membrane processes, the possibility of CDI fouling still exists due to contaminants in the effluent. CDI fouling can occur either in the flow path space or on the surfaces of ion exchange membranes and electrodes. In addition, the CDI deionization performance is sensitive to design and operating variables such as voltage, flow rate, sorption time, temperature, and electrode materials [30,31,32,33,34,35]. With so many variables involved, it is difficult to find the optimal conditions for CDI [36].

There are various approaches to optimize engineering systems and water treatment processes based on optimal algorithms, artificial intelligence, machine learning, the finite element method, and computational fluid dynamics [37,38,39,40,41,42,43]. The optimization could be implemented during the initial design assessment stage, leading to a reduction in cost and time [42,43]. Response surface methodology (RSM), which is one of such optimization methods, is efficient in deriving optimal conditions when various variable factors are present. RSM can predict and optimize results according to arbitrary varying levels in the desired range, reducing unnecessary experimental sets and conducting only the necessary experiments to derive optimal conditions [44]. Previous studies have used RSM to optimize treatment processes for municipal wastewater, industrial wastewater, landfill leachate, etc. [45].

The objective of this study was to apply CDI as an affordable technology for wastewater reclamation in CCPPs. The outline of this paper is as follows: First, the effect of operating variables on the TDS removal of CDI was investigated using the one-factor-at-a-time (OFAT) method. Second, further optimization of the operating variables for CDI was performed based on response surface methodology (RSM). Once the optimum conditions were obtained, long-term operation of the CDI was conducted to study performance degradation due to fouling. The foulants in the membranes, spacers, and electrodes were analyzed using FE-SEM, EDS, and XPS. Finally, the effectiveness of chemical cleaning was evaluated.

## 2. Materials and Methods

### 2.1. Materials

#### 2.1.1. Principle of CDI

As one of the CDI types, membrane-coupled capacitive deionization (MCDI) was used in this work [46,47,48,49,50]. Ion removal by CDI is performed by repeating the ion adsorption and desorption phases [46,47,48,49,50,51,52]. Water is demineralized in an adsorption phase and the electrodes are regenerated in a desorption phase [8,50,53]. MCDI combines an ion exchange membrane on the electrode surface, reducing the influence of ions of the same charge and reducing the possibility of electrode fouling, resulting in better deionization performance [25,54,55]. CDI in this paper refers to MCDI. The principle of CDI is shown in Figure 1.

#### 2.1.2. CDI Module

The lab-scale MCDI module consists of a few acrylic plates, a cathode, a cation exchange membrane, a spacer, an anion exchange membrane, and an anode (Figure 2). The cathode and an anode made of activated carbon with a size of W100 × L100 mm^2^ were purchased from Pureechem (Chungcheongbuk-do, South Korea). The adsorption capacity of the electrodes was 16 mg/g. The information about the pore volume and the surface area was not disclosed because the manufacturer considers it confidential information. The ion exchange membrane made of polyethylene with a size of W100 × L100 mm^2^, and the spacer made of polyethylene terephthalate with a size of W110 × L110 mm^2^ were provided by Pureechem (Chungcheongbuk-do, South Korea) (Table 1). The upper-left and center of the MCDI unit cell were punched at a size of 1 cm to allow the inflow and outflow water to flow, respectively. The top and bottom of the module had terminals (electrical contacts) that were connected to the electrodes to apply electricity. Since the CDI process was operated in constant voltage mode, the voltage was set to be constant while the current changed.

#### 2.1.3. Experimental Setup

Figure 3 shows a schematic of an experimental apparatus for the CDI process. A pump was used to supply feed water to the CDI module, and a power supply was used to apply voltage to the CDI electrode. The conductivity sensor was connected to a computer for real-time measurement and data collection. An electrode and ion exchange membrane were used after being immersed in ultrapure water for 2 h or more to remove contaminants from surface pores to secure an adsorption site. In addition, adsorption and desorption operations were performed for at least one cycle to achieve concentration equilibrium by an electrical force.

In the adsorption phase, the electrodes were charged to a given voltage, and ions of the opposite charge were attracted to the electrodes. In the desorption phase, the electrodes were switched to a negative potential, which caused the ions to be released from the electrodes, and the ions were released to the feed water passing through the cell. Accordingly, the product water was obtained in the adsorption phase and the concentrate (brine) was generated in the desorption phase.

When the TDS removal decreased to less than 70%, the CDI unit was cleaned in clean-in-place (CIP) mode. When the cleaning began, the remaining feed water was drained and the tank containing cleaning chemicals was connected to the CDI unit. The cleaning was performed in the order of acid (1 mM HCl) and alkali (1 mM NaOH solution), and then all chemicals in the process were drained and sufficient flushing was performed using deionized water.

#### 2.1.4. Feed Solution

Real wastewater effluent from a CCPP (referred to as CCPP ‘A’) in Korea was used as the feed water. Table 2 summarizes the wastewater quality and discharge standard. There are three types of wastewater from a CCPP: power generation wastewater, washing wastewater, and cooling wastewater, and the water quality (especially TDS) fluctuates with time depending on the mixing ratio of the wastewater streams. Therefore, two types of mixed wastewater samples were used in this work. To determine the optimum operating conditions, the high-TDS wastewater (referred to as F1) was used, which represents the worst-case scenario. To evaluate the long-term operating performance, the wastewater with low TDS (referred to F2) was used, which represents the normal operating conditions. The details of the water quality of the wastewater are summarized in Table 2. Since CDI was used to recycle the wastewater, the target water quality was set to the industrial water reuse standard rather than the discharge standard. Thus, the product water quality was compared to the reuse standard. The discharge standard and the industrial water reuse standard are also listed in Table 2.

### 2.2. Methods

#### 2.2.1. Application of RSM

The central composite design method was used to derive the optimal operating conditions of the CDI [56,57,58]. Voltage $\left(X_{1}\right)$, flow rate $\left(X_{2}\right)$, and desorption ratio $\left(X_{3}\right)$ were used as input variables (factor), and the TDS removal rate $\;(Y_{1})$ and production quantity $\;(Y_{2})$ were designed as output variables (response) using a commercial software (Design-Expert 12.0). Table 3 summarizes the factors and responses used in the RSM.

The conversion formula for TDS is given by Equation (1) [59]:(1)TDS(mg/L)=k×EC
where *k* is a correction coefficient and *EC* (µS/cm) is an electrical conductivity.

The formula of rejection (%) is given by Equation (2):(2)$$R\left(\%\right)=\frac{C_{0}-C}{C_{0}}\times 100\;$$
where $C_{0}$ is the initial concentration of the feed water, and C is the concentration of product water.

The production quantity (mL) of formula is given by Equation (3):(3)$$Q\left(mL\right)=V\times t_{AD}\times TC\;$$
where V is the flow rate (mL/min), $t_{AD}$ is the time of the adsorption time, and TC is the total cycle. The standard for 1 cycle was set as the sum of adsorption and desorption times, and the criterion of *TC* was 24 h.

The energy consumption (kWh/m^3^) of formula is given by Equation (4) [60]:(4)$$E=\frac{\int_{\Delta t}^{t_{AD}}IVdt-\eta \Sigma \int_{\Delta t}^{t_{DE}}IVdt}{\int_{\Delta t}^{t_{AD}}Qdt}$$
where IV is the current–voltage product (kW), *η* is the fraction of energy actually recovered and reused to power another charging phase, $t_{AD}$ is the time of the adsorption time (h), $t_{DE}$ is the time of the desorption time (h), and *Q* is the flow rate of the product water (m^3^/h).

Figure 3 shows a flow chart of the RSM for the optimization of the operating conditions for the CDI process. First, the objective of the optimization was clarified. The key variables, including factors (inputs) and responses (outputs), were then identified. The experimental design was then performed to determine the details of the experiments based on the central composite design (CCD) method. Using this experimental matrix, a series of experiments were conducted and the results were analyzed using the RSM regression method. In addition, ANOVA (analysis of variance) of the results was also performed to validate the regression models. When the ANOVA test failed, the experimental design was modified and the other steps were repeated. Otherwise, the regression model was considered valid. Using the regression model, optimal conditions were explored through both graphical and point optimization.

The R^2^ value for the regression models obtained from RSM trials was calculated using the Design-Expert software. The models were also validated using ANOVA. The larger the value of F and the smaller the value of *p*, the more significant the corresponding coefficient term [56]. The value of *p* was lower than 0.05, indicating that the model may be considered to be statistically significant [57].

#### 2.2.2. Field Emission Scanning Electron Microscopy (FE-SEM) and Energy Dispersive X-ray Spectrometer (EDS) Analysis

The foulants on the membrane surfaces, spacers, and the electrode were examined to provide detailed information on the morphology and composition of the fouling materials [61,62]. A field emission scanning electron microscope (FE-SEM) and an X-ray spectrometer (EDS) (JSM-7160F from JEOL LTD) were applied to each sample. Prior to analysis, the samples were completely dried and coated with platinum for 30–45 s.

#### 2.2.3. X-ray Photoelectron Spectrometer (XPS) Analysis

An X-ray photoelectron spectrometer Escalab250 (XPS) (Thermo Scientific) was used to analyze inorganic compounds, elements, etc. This technique has the advantage of being able to measure the elemental composition as well as the chemical and electronic state of the atoms within a material [63].

#### 2.2.4. Sample Analysis

The pH and conductivity data were collected in real time using a WTW Multi 3620 IDS. An inductively coupled plasma optical emission spectrometer (ICP-OES) (Agilent) and an ion chromatograph 5000+ (Thermo Scientific) were used for cation and anion analysis, respectively. Turbidity was determined using a 2100Q and TOC was analyzed using a Shimadzu TOC analyzer.

## 3. Results and Discussion

### 3.1. Effects of Operating Variables

Initially, the effect of voltage, flow rate, and adsorption/desorption ratio on TDS removal in the CDI process was investigated using the one-factor-at-a-time (OFAT) method. The F1 feed water was used for different operating conditions.

#### 3.1.1. Voltage

Figure 4 shows the effect of voltage on TDS removal. The voltage was applied in increments of 0.3 V and ranged from 1.2 V to 1.8 V. The TDS removal was highest at 86.3% at 1.5 V. In general, higher voltage can result in higher TDS removal. However, the TDS removal decreased at 1.8 V because the electrolysis of water occurs above 1.5 V, resulting in a decrease in ion adsorption efficiency.

#### 3.1.2. Flow Rate

The TDS removal at different flow rates is shown in Figure 5. The flow rate conditions were set in increments of 5 mL/min, ranging from 10 mL/min to 20 mL/min. As the flow rate increased, the TDS removal decreased. For example, the highest TDS removal (94.3%) was achieved at the highest flow rate of 20 mL/min. This is due to a decrease in contact time as the flow rate increases.

#### 3.1.3. Ratio of Sorption Time

The ratio of desorption time to adsorption time was adjusted from 1:0.8 to 1.2 in increments of 0.2, and the results are summarized in Figure 6. At the ratio of 1:1, the TDS removal was the highest (86.3%). When the desorption time was shorter than the adsorption time, the TDS removal decreased to 78.7%. This is because the ions adsorbed on the electrode are not sufficiently desorbed. When the desorption time was too long, the TDS removal was also low due to the occurrence of electrode aging, pore structure change, and by-product generation caused by the long desorption time.

### 3.2. Optimization of Capacitive Deionization in CCP under RSM

Although the OFAT approach is helpful for qualitatively understanding the influence of operating variables, it is inefficient for finding optimal conditions and is unable to reveal interactions among variables. Therefore, RSM was used to further optimize the operating variables for CDI. Table 4 shows the experimental design matrix and results from Design-Expert. Twenty-one experimental conditions were identified and a series of laboratory-scale experiments were conducted. The corresponding values for TDS removal (Response 1) and water production (Response 2) are also shown in Table 4.

The RSM results for the TDS removal and water production are shown in Table 5 and Table 6, respectively. The R^2^ values for several regression models, using linear, 2FI, quadratic, and cubic models, were compared. The sequential and lack-of-fit *p*-values were also calculated. The results indicated that the quadratic model was the most suitable for TDS removal $\;(Y_{1})$, and the 2FI model was suitable for water production $\;(Y_{2}$).

#### 3.2.1. TDS Removal

As shown in Table 4, the TDS removal varied from 47% to 91% by changing the voltage, flow rate, and adsorption/desorption ratio. Based on these results, a regression equation was derived as given by Equation (5):(5)$$Y_{1}=+84.47+12.26X_{1}-5.83X_{2}-5.93X_{3}+0.5162X_{1}X_{2}+2.81X_{1}X_{3}+1.37X_{2}X_{3}-4.96X_{1}{}^{2}-3.09X_{2}{}^{2}-4.97X_{3}{}^{2}$$

$Y_{1}$: TDS Removal

$X_{1}$: Voltage

X2: Flow rate

$X_{3}$: Sorption ratio

According to the ANOVA (analysis of variance) results (Table 7), the model F-value was 28.94 (*p*-value < 0.0001), which means that the model is significant. In addition, the lack-of-fit F-value was 2.31, so it can be concluded that the lack of fit is not significant. In addition, the coefficient of determination (*R^2^*) was 0.9595, which is also sufficiently high. This indicates that the regression equation can be used to predict the TDS removed.

The response surface and contour plots for TDS removal are shown in Figure 7, which allows the visualization of the effect of the input variables. Increasing the voltage increased the TDS removal, while increasing the flow rate decreased the TDS removal. Increasing the sorption ratio also resulted in decreased TDS removal. However, the dependence of TDS removal on these variables is not linear, indicating that there were interactions among the variables.

#### 3.2.2. Water Production

Equation (6) was obtained to predict water production using the data in Table 4:(6)$$Y_{2}=+10747.62+3843.90X_{2}-925.00X_{3}-725.00X_{2}X_{3}$$

$Y_{2}$: Water Production

X2: Flow rate

$X_{3}$: Sorption ratio

According to the results of ANOVA (analysis of variance) (Table 8), the F-value of the model was 248.67 (*p*-value < 0.0001), which means that the model is significant. The corresponding coefficient of determination *R^2^* was 0.9907. Contrary to the case of TDS removal, the voltage was found to have no effect on water production.

Figure 8 shows the response surface and contour plots for water production. An increase in flow rate resulted in an increase in water production, while an increase in sorption ratio resulted in a slight decrease in water production. As mentioned earlier, the water production was constant regardless of the voltage. As can be seen in the figure, the water production is almost linearly proportional to the flow rate.

#### 3.2.3. Graphical Optimization

There are two criteria to consider for CDI operation in this work. First, the TDS removal should be greater than 80%, which is required for wastewater reclamation. Second, the water production should be greater than 10,000 mL. To explore the conditions to satisfy both criteria at the same time, graphical optimization was carried out as shown in Figure 9. Here, the yellow areas correspond to the conditions for meeting the criteria. On the other hand, the gray areas correspond to conditions where the required criteria were not met. The desired ranges for voltage, flow rate and sorption ratio were then identified. A point optimization was also performed to determine the optimum point. The asterisks (star marks) indicate the proposed optimum conditions, which correspond to the voltage of 1.5 V, the flow rate of 15 mL/min and the sorption ratio of 1:0.8.

It should be noted that there are two major limitations to the RSM approach. The first is that RSM is a “black box” approach that lacks flexibility. If the model needs to be changed to describe something physically slightly different, a lot of work is required to develop a new model. The second limitation is that it is a local analysis. The response surface developed is invalid for regions other than the factor ranges studied. Accordingly, the RSM models developed here are limited to the conditions considered in this study.

#### 3.2.4. Results of Product Water Quality Analysis

The summary of the product water quality from the experiment under the optimum conditions (the voltage of 1.5V, the flow rate of 15 mL/min, and the sorption ratio of 1:0.8) is shown in Table 9. The TDS removal was 86.31%, which is 7.14% higher than the target removal. The water quality standard for industrial water reuse was also met.

### 3.3. Long-Term Operation of CDI

Although the previous experiments allow the determination of the optimum conditions for TDS removal and water production, they were carried out within a few hours. Therefore, a long-term (up to 10 days) continuous operation of the CDI was performed to study the fouling behavior and to identify the main foulants using the F2.

#### 3.3.1. Reduction in TDS Removal and Energy Consumption

The experiment was conducted on a laboratory scale for a total of 10 days on a 24-h basis. The average TDS removal decreased over time from 93.65% to 55.70% as shown in Figure 10. In addition to the average values, the maximum and minimum TDS removals are shown in Table 10.

Energy consumption was also calculated using equation (4) and the results are summarized in Table 10. Since the amount of water produced by the CDI process is constant (11.625 L/day), the denominator of equation (4) has a constant value regardless of time. On the other hand, the numerator of this equation is almost proportional to the amount of ions removed by the CDI, so it decreased over time due to a decrease in TDS removal. Accordingly, the calculated energy consumption, assuming no energy recovery, decreased from 3.51 kWh/m^3^ to 1.93 kWh/m^3^ during the 10-day operation. Assuming 100% energy recovery, it decreased from 0.62 kWh/m^3^ to 0.18 kWh/m^3^. This is undesirable because the quality of the product water deteriorated over time. To further investigate these phenomena, a series of analyses were performed on the membrane surface, spacers and electrodes.

#### 3.3.2. Results of SEM–EDS Analysis

SEM–EDS analysis was performed for the electrodes before and after the CDI experiments. Figure 11a shows the SEM images of the electrode before the experiment. The intact electrode was composed of particulate materials with porous structures. After the experiment, the cathode was found to be covered with foulants. This can be seen in Figure 11b. On the other hand, the anode did not have any significant fouling as shown in Figure 11c. The EDS spectra are consistent with the results of the SEM analysis. Compared with the EDS spectrum for the intact electrode (Figure 11d), the EDS spectrum for the used cathode (Figure 11e) shows the peaks for Ca (9.67%), Si (2.92%) and Mg (1.03%). On the contrary, the EDS spectrum for the used anode (Figure 11f) does not show many foreign elements except for small amounts of S (0.06%) and Cl (0.05%). These results suggest that the fouling of the electrode preferentially occurs on the cathode side. Since the cathode adsorbs positively charged ions such as Ca^2+^ and Mg^2+^, it can be covered by the deposits formed by these ions and dissolved organic matter [64].

The ion exchange membranes were also examined using the SEM–EDS technique. Before the CDI experiment, the surfaces of the cation exchange membrane (CEM) and anion exchange membrane (AEM) were clean as shown in Figure 12a,b. After the experiment, the surfaces were covered with gel-like and thread-like deposits as shown in Figure 12c,d. The analysis of the deposits by EDS showed that the deposits on the CEM and AEM were different. The intact CEM contained a small amount of S and Na and the intact AEM had a small amount of N and Cl (Figure 12e,f) due to their functional groups [65]. After the experiment, Ca (2.69%), N (2.23%), Mg (0.53%), and Si (0.12%) were mainly found on the used CEM (Figure 12g), while O (21.61%) and S (1.88%) were observed on the used AEM (Figure 12h).

The spacers were also observed by SEM–EDS. As shown in Figure 13a,b, the spacers were blocked by the deposits after the CDI experiment. Based on the morphology, they were assumed to be inorganic scales. The EDS analysis in Figure 13c,d confirmed this. It was found that the deposits in the used spacer contained Ca (4.79%), Mg (10.25%) and silica (1.95%).

#### 3.3.3. Results of XPS Analysis

In addition to EDS, XPS was used to further analyze the compositions of the deposits in the spacer. Figure 14 shows the XPS spectrum of the spacer after the CDI experiment. There were peaks corresponding to Mg 1s, O 1s, N 1s, Ca 2p, C 1s and Si 2p. Among them, the peaks for Mg, Ca, and O were higher than the others. These results are consistent with the previous results of the EDS analysis.

### 3.4. Cleaning

Previous results have shown that inorganic scale containing Ca and Mg is the main cause of CDI fouling. Since they degrade the treated water quality of CDI and reduce water production, it is necessary to remove them during or after CDI operation. Considering the compositions of the foulants, 1 mM HCl and 1 mM NaOH solutions were used as cleaning agents. The cleaning was performed by applying the acidic and alkaline solutions sequentially when the TDS removal decreased to less than 70%. Figure 15 shows the changes in TDS removal by CDI with time and the effectiveness of cleaning. The average TDS removal was 93.65% on day 1 and decreased to 68.82% on day 7.

The average TDS removal was 93.65% on day 1 and decreased to 68.82% on day 7. One of the reasons for this degradation is the electro-oxidation in electrodes. Due to the operation of the cell at a high voltage of 1.5 V, the anode can be electro-oxidized, leading to a reduction in capacity via a loss of pore structure and changes in surface chemistry. The elemental analysis indicated an increase in oxygen content, which may be associated with the electro-oxidation. Nevertheless, it should be noted that the TDS removal was restored to the initial level of 93.02% after the chemical cleaning. If electro-oxidation occurs, then the changes in the electrode should be irreversible and cannot be reverted by the chemical cleaning. Accordingly, further works will be required to clearly investigate the effect of electro-oxidation on the CDI performance degradation. Moreover, it is also necessary to optimize the cleaning methods for the long-term operation of CDI in future studies.

## 4. Conclusions

In this study, the CDI process has been proposed for the treatment of wastewater from CCPPs. The effect of operating conditions such as voltage, flow rate, and adsorption/desorption time ratio on the TDS removal in CDI was analyzed using both OFAT and RSM approaches. At the optimum conditions (the voltage of 1.5 V, the flow rate of 15mL/min, and the sorption ratio of 1:0.8), the TDS removal was 86.31%, which meets the requirement for industrial water reuse. However, a long-term CDI test showed a significant decrease in TDS removal from 93.65% to 55.70% over 10 days. To identify the foulants, the anodes, cathodes, CEMs, AEMs, and spacers were analyzed using SEM–EDS and XPS. It was found that fouling did not occur on the anode but on the cathode. Foulants were also observed in the CEMs, AEMs and spacers and they mainly contained Ca, Mg and Si. Accordingly, inorganic foulants, which may be associated with the organic matter, seemed to be responsible for the deterioration of the CDI performance. After the CDI test, chemical cleaning was performed using the acid and alkali solutions, which could recover the TDS removal from 68.82% to 92.03%. This indicates that CDI fouling can be controlled by chemical cleaning.

The significance of this paper lies in the application of the CDI process to the reclamation of CCPP wastewater, which is currently discharged despite its potential for reuse. Accordingly, this paper can contribute to the reduction of water withdrawal in CCPPs by accelerating the adoption of wastewater reclamation by the CDI process. One of the findings in this paper is the occurrence of fouling during the long-term operation of the CDI process, which is attributed to inorganic scale and organic matter. Although a chemical cleaning procedure has been proposed in this study, it has not yet been fully optimized. Therefore, further work will be required to explore the optimal methods and conditions for mitigating CDI fouling. It will also be necessary to develop pre-treatment techniques for this CDI process.

## Figures and Tables

**Figure 1 membranes-13-00316-f001:**
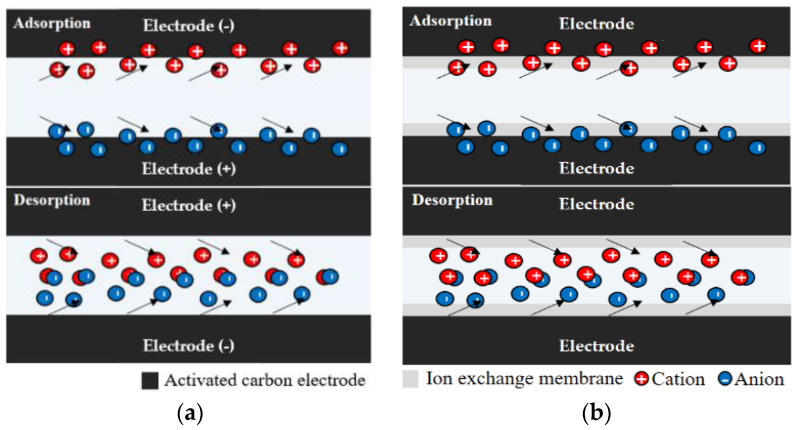
Principle of (**a**) capacitive deionization (CDI); (**b**) membrane capacitive deionization (MCDI).

**Figure 2 membranes-13-00316-f002:**
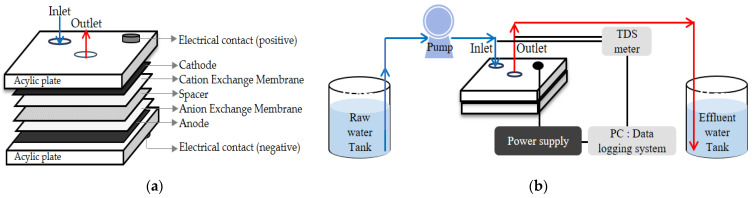
(**a**) Configuration of the lab-scale MCDI module unit cell; (**b**) schematic diagram of the lab-scale experimental setup for CDI process.

**Figure 3 membranes-13-00316-f003:**
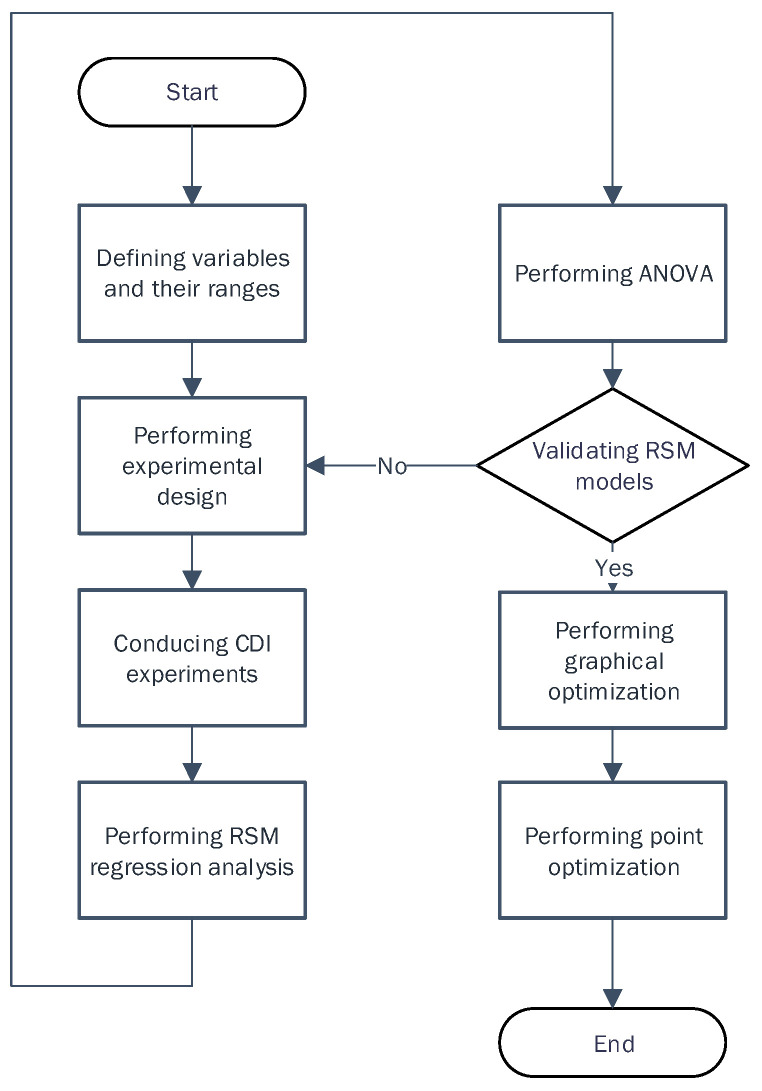
Flow chart for the RSM-based optimization.

**Figure 4 membranes-13-00316-f004:**
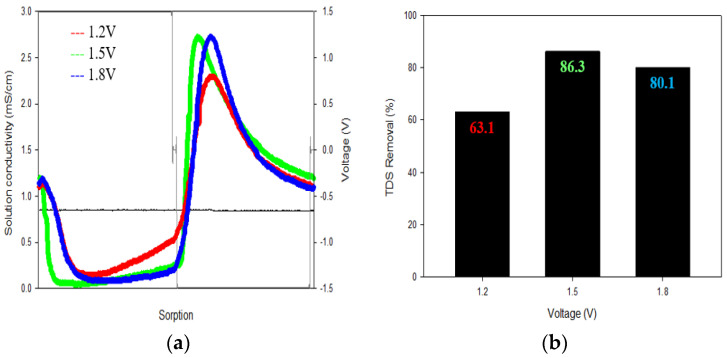
The results of the applied voltage of the CDI: (**a**) Voltage was set in 0.3 V increments from 1.2 V to 1.8 V; (**b**) The TDS removal rate was the highest at 86.3% at 1.5 V.

**Figure 5 membranes-13-00316-f005:**
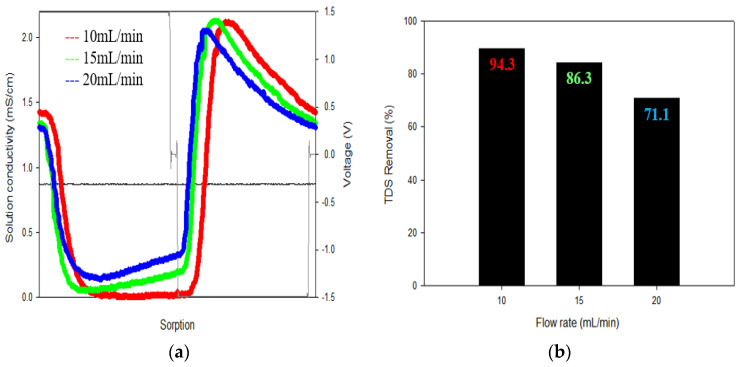
The results of the flow rate of the CDI: (**a**) Flow rate was set in 5 mL/min increments from 10 mL/min to 20 mL/min; (**b**) The TDS removal rate was the highest at 94.3% at 10 mL/min.

**Figure 6 membranes-13-00316-f006:**
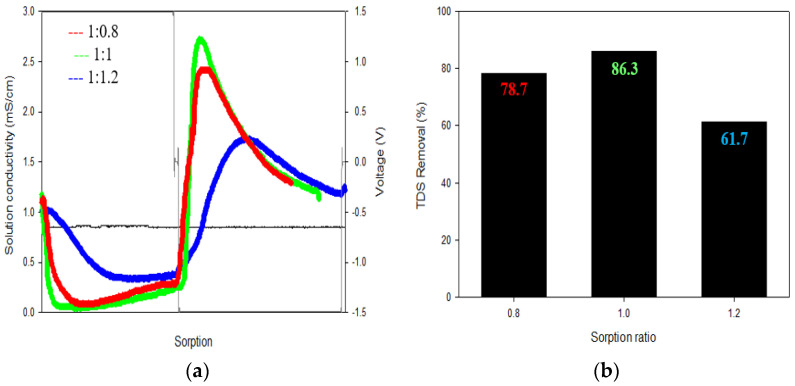
The results of the sorption ratio (adsorption/desorption) of the CDI: (**a**) Sorption ratio was set at 1:0.8, 1:1 and 1:1.2; (**b**) The TDS removal rate was the highest at 86.3% at 1:1.

**Figure 7 membranes-13-00316-f007:**
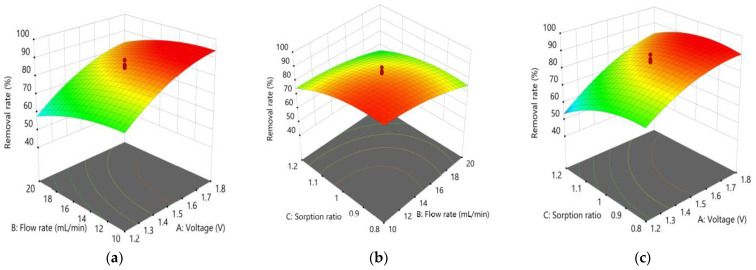
3D response surface plots showing the effects of the three main factors on TDS removal using the CDI: (**a**) voltage and flow rate; (**b**) flow rate and sorption ratio; (**c**) voltage and sorption ratio.

**Figure 8 membranes-13-00316-f008:**
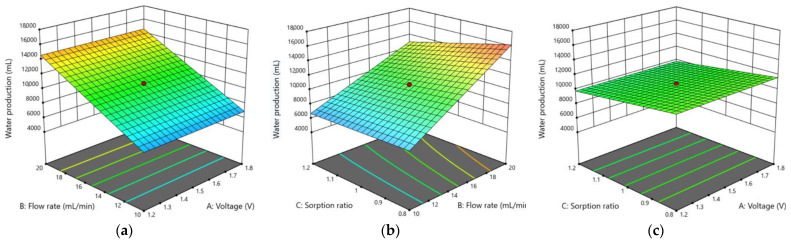
3D response surface plots showing the effects of the three main factors on water production using the CDI: (**a**) voltage and flow rate; (**b**) flow rate and sorption ratio; (**c**) voltage and sorption ratio.

**Figure 9 membranes-13-00316-f009:**
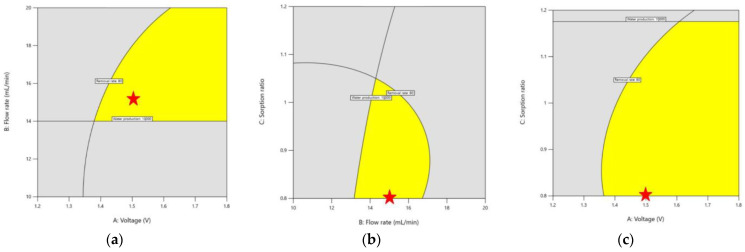
Contour overlay plots showing the graphical optimization of the CDI (yellow area: conditions for meeting the water reuse criteria, grey area: conditions failing to meet the criteria, star marks (asterisks): proposed optimum conditions). (**a**) TDS removal vs. voltage and flow rate; (**b**) TDS removal vs. voltage and sorption ratio; (**c**) TDS removal vs. flow rate and sorption ratio.

**Figure 10 membranes-13-00316-f010:**
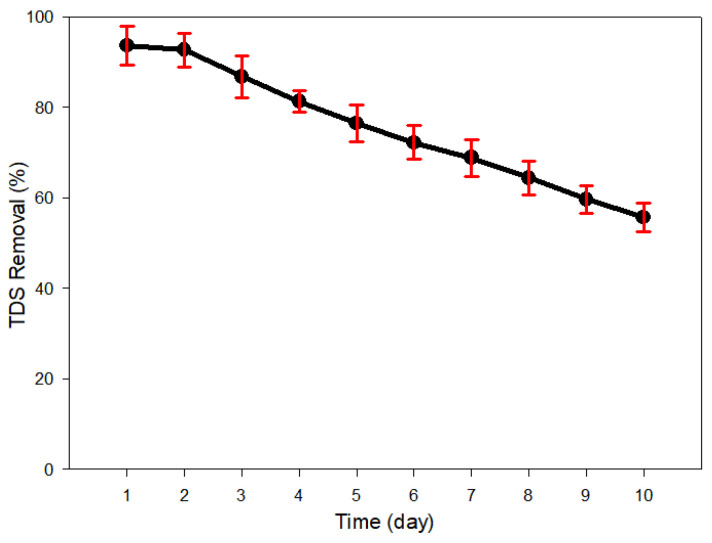
The variation of TDS removal rate over 10 days of 24 h per day operation. The average TDS removal rate was 93.65% on Day 1 and gradually fell to 55.70% on Day 10.

**Figure 11 membranes-13-00316-f011:**
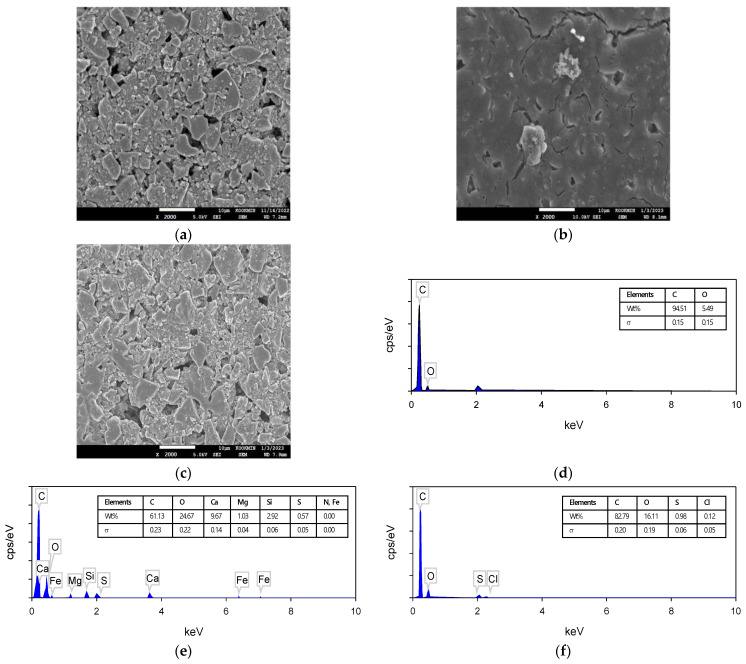
FE-SEM images and EDS spectrum of electrodes surface before and after use: (**a**) virgin electrodes for different times; (**b**) cathode fouled by Ca, Si, and Mg; (**c**) anode fouled by S and Cl at 2000 times, respectively; (**d**–**f**) show the EDS spectra of (**a**–**c**), respectively.

**Figure 12 membranes-13-00316-f012:**
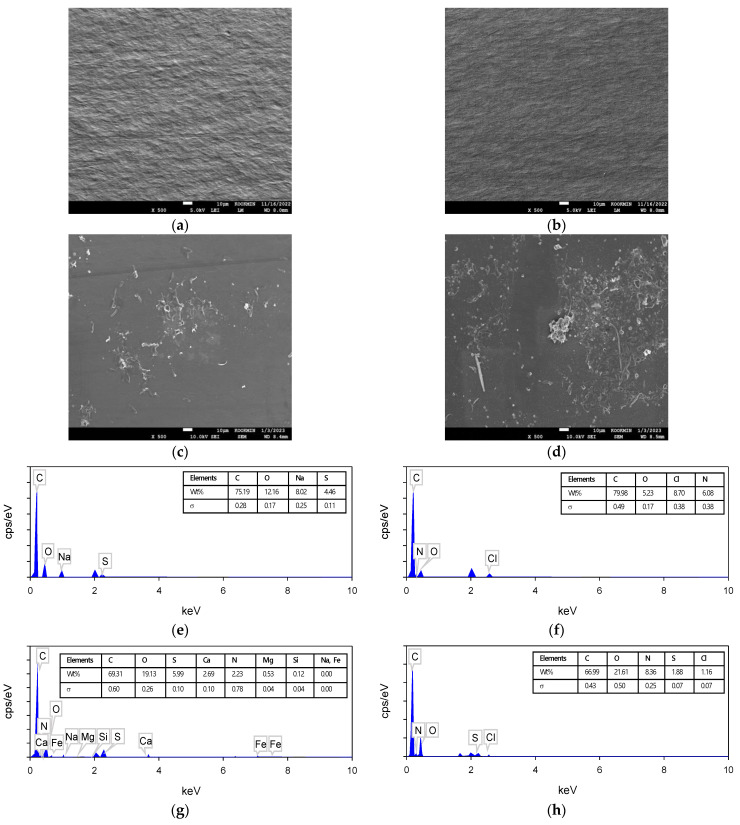
FE-SEM images and EDS spectrum of the ion exchange membrane surface before and after use: (**a**) and (**b**) virgin cation and anion exchange membrane; (**c**) CEM fouled by Ca, N, Mg, Si and S; (**d**) AEM fouled by S at 500 times, respectively; (**e**–**h**) show the EDS spectra of (**a**–**d**), respectively.

**Figure 13 membranes-13-00316-f013:**
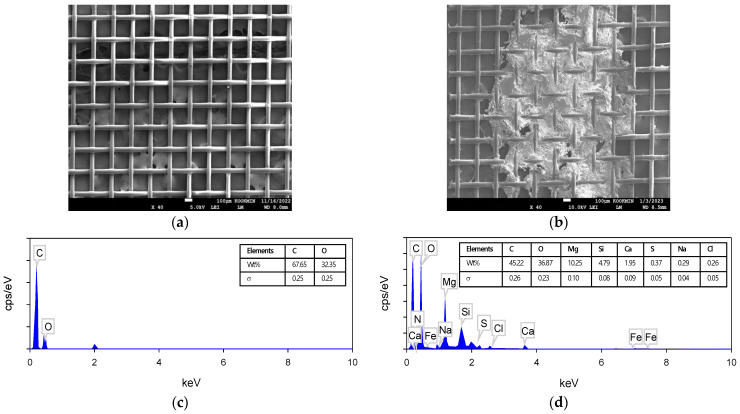
FE-SEM images and EDS spectrum of the spacer surface before and after use: (**a**) virgin spacer; (**b**) spacer fouled by Ca, Mg and Si at 40 times, respectively; (**c,d**) show the EDS spectra of (**a,b**), respectively.

**Figure 14 membranes-13-00316-f014:**
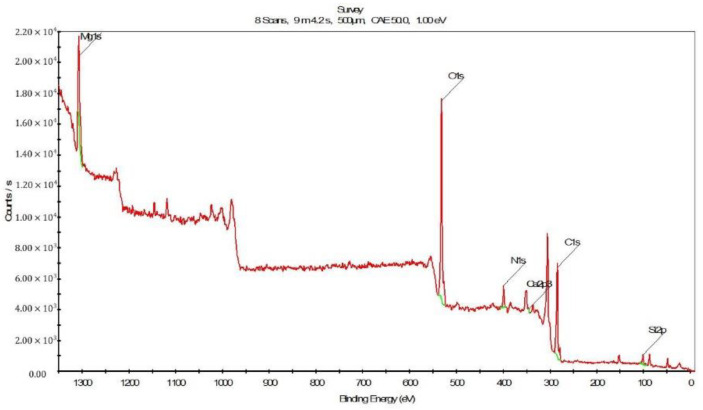
The XPS spectra of the deposits in the spacer.

**Figure 15 membranes-13-00316-f015:**
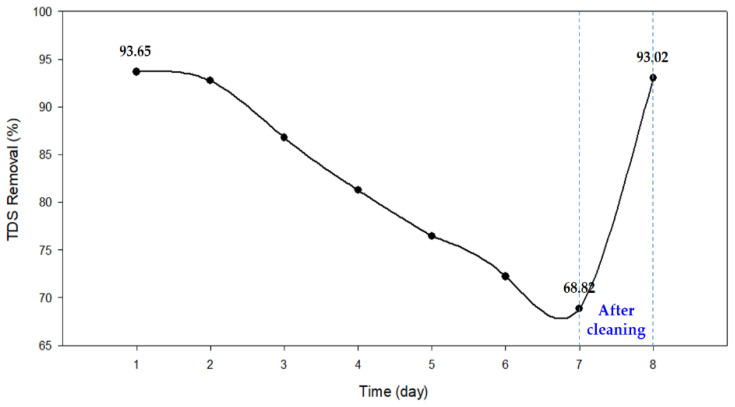
The variation of TDS removal rate over 7 days of 24 h per day operation and after cleaning at 8 days; The average TDS removal rate was 93.65% on day 1 and gradually fell to 68.82% on day 7. After cleaning processes, the TDS removal rate recovered to the initial level of 93.02%.

**Table 1 membranes-13-00316-t001:** Summary of MCDI module specification.

	Material	Standard	Capacity
Electrode	Activated Carbon, Graphite	W100 × L100 × T0.6 mm^3^	16 mg/g or more
CEM	Polyethylene	W100 × L100 × T0.015 mm^3^	1.6 meq/g or more (Sheet resistance: 0.5 Ω cm^2^ or under)
AEM	Polyethylene	W100 × L100 × T0.015 mm^3^	1.7 meq/g or more (Sheet resistance: 0.3 Ω cm^2^ or under)
Spacer	Polyethylene terephthalate	W110 × L110 × T0.01 mm^3^	-

**Table 2 membranes-13-00316-t002:** The water quality of the wastewater and standards of industrial discharge limits.

Items	Unit	Feed Conditions		Standard to Discharge	Target to Removal Rate (%)	
		F1	F2		F1	F2
pH	-	8.29	7.3	6~8	-	-
Conductivity	μS/cm	2402	1055	<500:reuse	79.17	52.61
TOC	mg/L	3.2	8.7	<25	-	-
Cl^−^	mg/L	145	123	< 7	95.17	94.31
Alkalinity (as CaCO_3_)	mg/L	105	77	<10	90.48	87.01
Hardness (as CaCO_3_)	mg/L	267	285	<10	96.25	96.49
Turbidity	NTU	0.12	0.94	<2	-	-
Fe	mg/L	0.017	0.601	<0.05	-	91.68
NH^4+^	mg/L	0.03	0.1	<0.03	-	70.00
SO_4_^2−^	mg/L	85	74	<8	90.59	89.19

**Table 3 membranes-13-00316-t003:** Factors and responses used in RSM.

Name	Units	Type	Values
Voltage	V	Factor	1.2, 1.5, 1.8
Flow rate	mL/min	Factor	10, 15, 20
Sorption ratio (ad:de)	-	Factor	1.0:0.8, 1.0:1.0, 1.0:1.2
TDS Removal rate	%	Response	
Water Production	mL	Response	

**Table 4 membranes-13-00316-t004:** Arrangement of response surface design.

Std	Run	Factor 1	Factor 2	Factor 3	Response 1	Response 2
		*X_1_*: Voltage (V)	*X_2_*: Flow Rate (mL/min)	*X_3_*: Sorption Ratio	*Y_1_*: TDS Removal (%)	*Y_2_*: Water Production (mL)
3	1	1.2	20	0.8	57.47	16,000
12	2	1.5	15	1	88.91	10,800
11	3	1.5	15	1	85.24	10,800
9	4	1.5	15	1	84.78	10,800
10	5	1.5	15	1	86.31	10,800
5	6	1.2	10	1.2	56.17	6550
6	7	1.8	10	1.2	89.41	6550
13	8	1.5	15	1	85.44	10,800
8	9	1.8	20	1.2	79.04	13,100
7	10	1.2	20	1.2	49.87	13,100
4	11	1.8	20	0.8	81.53	16,000
2	12	1.8	10	0.8	91.23	6550
1	13	1.2	10	0.8	75.37	6550
19	14	1.2	15	1.4	47.60	9000
17	15	1.5	23	1	64.83	16,560
20	16	1.5	15	1	80.19	10,800
21	17	1.5	15	1	80.44	10,800
14	18	1	15	1	49.99	10,800
18	19	1.5	15	0.6	79.47	13,500
15	20	2	15	1	88.32	10,800
16	21	1.5	7	1	84.95	5040

**Table 5 membranes-13-00316-t005:** Summary of quadratic model on the response $\;(Y_{1})$: TDS removal.

Source	Sequential *p*-Value	Lack of Fit *p*-Value	Adjusted R^2^	Predicted R^2^	
Linear	<0.0001	0.0042	0.6606	0.5631	
2FI	0.8076	0.0024	0.6147	0.4262	
Quadratic	<0.0001	0.1686	0.9263	0.7414	Suggested
Cubic	0.1982	0.1903	0.9461	−0.1471	

**Table 6 membranes-13-00316-t006:** Summary of 2FI model on the response $\;(Y_{2})$: water production.

Source	Sequential *p*-value	Lack of Fit *p*-value	Adjusted R^2^	Predicted R^2^	
Linear	<0.0001		0.9659	0.9465	
2FI	0.0009		0.9867	0.9604	Suggested
Quadratic	0.4543		0.9865	0.9397	
Cubic	0.0406		0.9941	0.5363	

**Table 7 membranes-13-00316-t007:** ANOVA for the response surface quadratic model.

Source	Sum of Squares	df	Mean Square	F-Value	*p*-Value	
Model	4107.27	9	456.36	28.94	<0.0001	Significant
$X_{1}$: Voltage	2038.05	1	2038.05	129.26	<0.0001	
$X_{2}$: Flow rate	445.61	1	445.61	28.26	0.0002	
$X_{3}$: Sorption ratio	562.28	1	562.28	35.66	<0.0001	
$X_{1}X_{2}$	2.13	1	2.13	0.1352	0.7201	
$X_{1}X_{3}$	63.23	1	63.23	4.01	0.0705	
$X_{2}X_{3}$	14.93	1	14.93	0.9471	0.3514	
$X_{1}{}^{2}$	356.01	1	356.01	22.58	0.0006	
$X_{2}{}^{2}$	121.93	1	121.93	7.73	0.0179	
$X_{3}{}^{2}$	669.14	1	669.14	42.44	<0.0001	
Residual	173.44	11	15.77			
Lack of fit	114.15	5	22.83	2.31	0.1686	Not significant
Pure Error	59.29	6	9.88			
Cor Total	4280.71	20				

**Table 8 membranes-13-00316-t008:** ANOVA for the response surface 2FI model.

Source	Sum of Squares	df	Mean Square	F-Value	*p*-Value	
Model	2.118 × 10^8^	6	3.529 × 10^7^	248.67	<0.0001	Significant
$X_{1}$: Voltage	0.0000	1	0.0000	0.0000	1.0000	
$X_{2}$: Flow rate	1.939 × 10^8^	1	1.939 × 10^8^	1365.94	<0.0001	
$X_{3}$: Sorption ratio	1.369 × 10^7^	1	1.369 × 10^7^	96.46	<0.0001	
$X_{1}X_{2}$	0.0000	1	0.0000	0.0000	1.0000	
$X_{1}X_{3}$	0.0000	1	0.0000	0.0000	1.0000	
$X_{2}X_{3}$	4.205 × 10^6^	1	4.205 × 10^6^	29.63	<0.0001	
Residual	1.987 × 10^6^	14	1.419 × 10^5^			
Lack of fit	1.987 × 10^6^	8	2.484 × 10^5^			
Pure Error	0.0000	6	0.0000			
Cor Total	2.137 × 10^8^	20				

**Table 9 membranes-13-00316-t009:** Summary of the product water quality and standards of industrial discharge limits using feed solution F1.

Items	Unit	Feed Conditions	Standard to Discharge	Product Water Quality of CDI
		F1		
pH	-	8.29	6~8	7.1
Conductivity	μS/cm	2402	<500:reuse	329
TOC	mg/L	3.2	<25	1.5
Cl^−^	mg/L	145	<7	13.8 (±1)
Alkalinity (as CaCO_3_)	mg/L	105	<10	7 (±1)
Hardness (as CaCO_3_)	mg/L	267	<10	37 (±1)
Turbidity	NTU	0.12	<2	0.23 (±1)
Fe	mg/L	0.017	<0.05	0.019 (±0.005)
NH^4+^	mg/L	0.03	<0.03	0.01 (±0.005)
SO_4_^2−^	mg/L	85	<8	7.7 (±1)

**Table 10 membranes-13-00316-t010:** Variations of TDS removal and energy consumption with time.

Time (day)	Removal (%)			Energy Consumption (kWh/m^3^)	
	Average	Maximum	Minimum	No Energy Recovery (*η* = 0)	100% Energy Recovery (*η* = 1)
1	93.65	97.27	88.64	3.51	0.62
2	92.72	95.73	88.27	3.38	0.46
3	86.77	91.45	82.15	3.13	0.37
4	81.27	83.73	78.91	2.88	0.31
5	76.44	80.82	72.64	2.68	0.27
6	72.20	75.82	68.36	2.55	0.27
7	68.82	73.36	65.36	2.41	0.25
8	64.41	68.73	61.18	2.28	0.23
9	59.66	68.18	57.00	2.09	0.20
10	55.70	59.45	53.18	1.93	0.18

## Data Availability

Not applicable.

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
