# Peer review of "Optimization and Evaluation for the Capacitive Deionization Process of Wastewater Reuse in Combined Cycle Power Plants"

_membranes, 2023, doi:10.3390/membranes13030316_

Round 1

Reviewer 1 Report

Please read the attachment. Thank you.

Reviewer 2 Report

Summary and General Comments

In this study, the authors used activated carbon electrodes for capacitative deionisation for the treatment of power plant wastewater.

Overall, the work is interesting to read and scientifically sound. The language is easy to understand, and the result and discussion support the conclusion. However, the work can be further improved. Please, see the specific comments below for details.

Specific Comments

1.   I do not think abbreviation CDI and CCPP are required in the title

2.   Abstract: I suggest the authors to include information about the electrode materials.

3.   Abstract: The abbreviation ad:de is not required as this term only appear once.

4.   In the Introduction, I suggest the authors to describe the problem of the CCPP wastewater, and how it can deteriorate the system.

5.   Some sections in the methodology are not clear, although it becomes obvious as readers proceed to the next section. I suggested the authors to emphasise that activated carbon is used as the main active agents and electrode for the removal of the ions at Lines 91-95, and activated carbon are used as both the cathode and anode?

6.   The authors should also specify the source of the activated carbon, whether it is made by the authors or purchased. The authors should also give general information such as the pore volume and surface area.

7.   The general setup of the whole experiment is quite clear, however, the section related to the CDI module is not clear. How is the wiring connected to the CDI module. For the power supply, the voltage is mentioned; however, the applied current is not. If the current is also controlled, it should be mentioned in the Methods section.

8.   The adsorption phase and desorption phase should be clearly explained in the methodology. It is not clear how exactly the desorption phase was carried out. Was ultrapure water used? All of this information should be included in the methodology.

9.   Tables 5-6. It is not clear what “suggested aliased” means.

10.             Fig. 9. The authors need to explain the grey versus yellow colouration and the red dot and star symbol in the caption of the figure.

11.             The cleaning procedure of section 3.4 should be mentioned in the methodology. Current version is not clear. Was acid and based pump into the CDI module or the CDI module needs to be taken apart and clean the individual parts?

Reviewer 3 Report

This work is an experimental study of the use of membrane capacitive deionization to purify water in combined cycle power plants. The main novelty is the use of such water streams, which to my knowledge data on CDI for such a stream is not present in the literature. Thus, overall this dataset will be of use for the field, but major deficiencies in the manuscript are still present, mainly:

- Missing a lot of relevant literature citations with regards to CDI generally, CDI experimental methods and CDI results which are pertinent to this study. For example, recent reviews and papers cover the topic of ion selectivity, which is relevant to this paper as it deals with mixed ion solutions with water quality dependent on which ion is removed:

Gamaethiralalage, J. G., et al. "Recent advances in ion selectivity with capacitive deionization." Energy & Environmental Science 14.3 (2021): 1095-1120.

Guyes, Eric N., et al. "Long-lasting, monovalent-selective capacitive deionization electrodes." npj Clean Water 4.1 (2021): 22.

Further, a study on the methods of reporting CDI data, which should be read by the authors (more on this below)

Hawks, Steven A., et al. "Performance metrics for the objective assessment of capacitive deionization systems." Water research 152 (2019): 126-137.

Finally a few studies on CDI degradation, which is relevant here as the authors also observe degradation (more on this below)

Uwayid, Rana, et al. "Characterizing and mitigating the degradation of oxidized cathodes during capacitive deionization cycling." Carbon 173 (2021): 1105-1114.

Algurainy, Yazeed, and Douglas F. Call. "Improving long-term anode stability in capacitive deionization using asymmetric electrode mass ratios." ACS ES&T Engineering 2.1 (2021): 129-139.

- The authors detect some severe degradation of their cell (loss of ~50% capacity in about 10 days). However, the one potentially important reason for this degradation is not stated by the authors, which is anode electro-oxidation. Due to operation of their cell at a high voltage of 1.5 V, the anode can electrooxidize causing loss of pore structure and changes in surface chemistry which reduce capacity. This is the main reported degradation mechanism is CDI. Electrooxidation is suggested by the authors elemental analysis of their electrodes post-experiment, which show large increases in electrode oxygen content.

- It would be relevant to report energy consumption data, for example in a manner described in the Hawks et al. paper provided previously in this report. 

- Some minor comments: Figure quality should be improved. Figure 10 looks "squished" and low resolution. Figure 11 EDS data is hard to read due to background color, etc.

Round 2

Reviewer 2 Report

After the revision, there are a lot of improvements and the methodology is more comprehensive now. I recommend this manuscript for publication in this journal.

Reviewer 3 Report

The authors have responded effectively to my comments. I have no further comments